# OpenReview forum: "A Deep Dive into Dataset Imbalance and Bias in Face Identification"
_NeurIPS.cc/2022/Workshop/TSRML — TSRML2022_

### Official Review · Reviewer_kq3K · 2022-10-17

**Overall Rating:** 5

**Summary:**

This paper studies the influence of different level data imbalance on face identification models. This paper does several empirical experiments to compare the bias in identifying male and female cause by dataset imbalance.

**Strengths:**

* The paper empirically compares the influence of imbalance dataset from different aspects, e.g. different number of images within one identity, different number of identities in each group, etc
* The results related to comparison between machines and human is really interesting


**Weaknesses:**

* The study is too limited on female and male group comparison and can be much improved with more detailed experiments among different groups.
* This paper is too limited on empirical study and can be improved by connecting with related papers on analyzing neural networks on fairness
* The comparison is only done on models trained with vanilla training and it can be more interesting by comparing the results from different training methods, especially the methods aimed to improve model fairness.

**Overall Recommendation:**

In all, I think the paper did some interesting empirical experiments on the bias caused by an imbalanced dataset. But the conclusions and experiments are weakened by limited groups, training methods and lack of theoretical analysis and discussion.

**Review Confidence:**

4: The reviewer is confident but not absolutely certain that the evaluation is correct

---

### Official Review · Reviewer_JJu3 · 2022-10-20
**The paper explores the effect of different kinds of data imbalance in Face Identification Task**

**Overall Rating:** 7

**Summary:**

This work explores the effect of different kinds of data imbalance of gender characteristic in face identification datasets. Specifically, imbalance in number of identities and images per identity are explored. The paper also showcases the phenomenon where inspite of data-imbalance, a model may appear un-biased. Lastly, the paper compares the biases in model with human biases.

**Strengths:**

- The paper addresses an important gap in research where facial identification systems are not evaluated on data-imbalance issues.
- The problem and experimental setup are well described
- The finding that randomly initialized models have higher male performance is very interesting

**Weaknesses:**

- The paper often makes observations such as 'we observe a decrease in performance
for both demographic groups when the images of that group constitute more than 60% of train data' (line 118) and  'The “fair point" where female accuracy is closest to male accuracy occurs when around 20% of images are of males' (line 122), but no attempt is made to understand what characteristics of the dataset contribute to these exact proportions. This makes it very hard to gain anything useful from these observations.


**Overall Recommendation:**

I'd recommend acceptance as the paper fills an important gap in research where facial identification systems are not evaluated on data-imbalance issues.

**Review Confidence:**

3: The reviewer is fairly confident that the evaluation is correct

---

### Official Review · Reviewer_vjMf · 2022-10-20
**Interesting problem, but weak analysis into different dimensions.**

**Overall Rating:** 6

**Summary:**

The paper attempts to address the dataset imbalance issue in Face Identification. The paper focuses on imbalance w.r.t. gender. With experiments with different proportions of the dataset (gender), the paper concludes that common machine learning techniques to create train and test splits can lead to Simpson’s paradoxes which lead to a false belief that a model is unbiased. The paper concludes with several actionable insights.

**Strengths:**

+ interesting observation in how the gender proportion affects face identification.

**Weaknesses:**

- only focused on gender, what about other aspects like race and etc?
- problem seems to be oversimplified to just gender, which maybe just looking at one dimension in a multivariate problem.
- more analysis beyond accuracy may help gain more insights into the problem.

**Overall Recommendation:**

The paper presents an observation on how the different proportions and imbalance of the datasets can affect face identification. The paper seems to take a rather simplified view of a very complex bias issue in the face identification. Also, it does miss out on further analysis of the problem beyond its affect on the accuracy. For example, further analysis into how per-class accuracy is affected, or how it is for different aspects of the dataset like gender would uncover more interesting insights.

However, I believe the paper makes a good effort to introduce an interesting issue and makes a cogent argument in its scope. Overall, the paper will be a good addition to the program.

**Review Confidence:**

4: The reviewer is confident but not absolutely certain that the evaluation is correct

---

### Decision · Program_Chairs · 2022-10-23

Accept